# Head-to-Head Comparison of BRAF/MEK Inhibitor Combinations Proposes Superiority of Encorafenib Plus Trametinib in Melanoma

**DOI:** 10.3390/cancers14194930

**Published:** 2022-10-08

**Authors:** Alexander Schulz, Jennifer Raetz, Paula C. Karitzky, Lisa Dinter, Julia K. Tietze, Isabell Kolbe, Theresa Käubler, Bertold Renner, Stefan Beissert, Friedegund Meier, Dana Westphal

**Affiliations:** 1Department of Dermatology, Faculty of Medicine and University Hospital Carl Gustav Carus, Technische Universität (TU) Dresden, Fetscherstrasse 74, 01307 Dresden, Germany; 2National Center for Tumor Diseases (NCT), 01307 Dresden, Germany; German Cancer Research Center (DKFZ), 69120 Heidelberg, Germany; Faculty of Medicine and University Hospital Carl Gustav Carus at TU Dresden, 01307 Dresden, Germany; Helmholtz-Zentrum Dresden–Rossendorf (HZDR), 01328 Dresden, Germany; 3Clinic and Polyclinic for Dermatology and Venereology, University Medical Center Rostock, 18055 Rostock, Germany; 4Institute of Clinical Pharmacology, Faculty of Medicine, TU Dresden, 01307 Dresden, Germany; 5Skin Cancer Center at the University Cancer Center (UCC) Dresden, University Hospital Carl Gustav Carus at TU Dresden, 01307 Dresden, Germany

**Keywords:** melanoma, resistance, BRAF, MEK, encorafenib, vemurafenib, dabrafenib, binimetinib, cobimetinib, trametinib

## Abstract

**Simple Summary:**

A decade ago, the diagnosis of metastatic melanoma was mostly a death sentence. This has changed since new therapies became widely available in the clinical setting. In addition to checkpoint inhibitors, targeted therapy with BRAF and MEK inhibitors is standard care for BRAF-mutated melanoma, which accounts for almost half of all melanoma cases. The second largest group of melanoma patients, whose tumors harbor a mutation in the NRAS gene, demonstrates only a limited response to targeted therapy with MEK inhibitors. The aim of this investigation was to directly compare all possible BRAF/MEK inhibitor combinations in addition to the currently applied regimens. The analyzed data suggested that the combination of the BRAF inhibitor encorafenib and the MEK inhibitor trametinib demonstrated the highest anti-tumor activity in both, BRAF- and NRAS-mutated melanoma. This combination is not presently used in patient treatment, and therefore, deserves an opportunity to become part of clinical trials.

**Abstract:**

BRAFV600 mutations in melanoma are targeted with mutation-specific BRAF inhibitors in combination with MEK inhibitors, which have significantly increased overall survival, but eventually lead to resistance in most cases. Additionally, targeted therapy for patients with NRASmutant melanoma is difficult. Our own studies showed that BRAF inhibitors amplify the effects of MEK inhibitors in NRASmutant melanoma. This study aimed at identifying a BRAF and MEK inhibitor combination with superior anti-tumor activity to the three currently approved combinations. We, thus, assessed anti-proliferative and pro-apoptotic activities of all nine as well as resistance-delaying capabilities of the three approved inhibitor combinations in a head-to-head comparison in vitro. The unconventional combination encorafenib/trametinib displayed the highest activity to suppress proliferation and induce apoptosis, acting in an additive manner in BRAFmutant and in a synergistic manner in NRASmutant melanoma cells. Correlating with current clinical studies of approved inhibitor combinations, encorafenib/binimetinib prolonged the time to resistance most efficiently in BRAFmutant cells. Conversely, NRASmutant cells needed the longest time to establish resistance when treated with dabrafenib/trametinib. Together, our data indicate that the most effective combination might not be currently used in clinical settings and could lead to improved overall responses.

## 1. Introduction

Cutaneous melanoma occurs from neoplastic changes in melanocytes and is frequently associated with major driver mutations such as BRAF (40–50%) or NRAS (20–30%) [1,2,3]. Since the development of first generation BRAF inhibitors (BRAFi), vemurafenib in 2011 and dabrafenib in 2013, melanoma patients with BRAFV600 mutations highly benefited from this targeted therapy (TT). More than 90% of patients demonstrated tumor control and 50–60% experienced partial or complete remission [4]. However, despite the remarkable rate of responders, the majority of patients developed resistance during the course of treatment, resulting in tumor progression [5]. A total of 15–20% of patients display primary (intrinsic) resistance and approximately 50% of patients treated with dabrafenib and trametinib acquired secondary resistance after 12 months [6,7,8]. Mechanisms for primary resistance include loss of tumor suppressor NF-1, mutation of RAC1, COT overexpression, dysregulation of receptor tyrosine kinase signaling, or loss of PTEN [9,10,11,12,13,14,15,16,17,18,19]. On the other hand, secondary resistance mechanisms to BRAFi involve alternative splicing or amplification of BRAF as well as an overexpression of CRAF or mutation of ARAF [20,21,22,23]. Activating mutations of NRAS, MEK1/2, or AKT1/3 have also been described to contribute to acquired resistance [19,24,25]. In order to delay the onset of resistance, the concomitant administration of BRAFi with MEK inhibitors (MEKi), binimetinib, trametinib or cobimetinib, is currently standard care for advanced BRAFV600-mutated (BRAF^mut^) melanoma and has also reduced secondary skin malignancies [26,27,28,29,30,31]. To date, the three approved BRAFi/MEKi combinations are encorafenib/binimetinib, dabrafenib/trametinib, and vemurafenib/cobimetinib.

For BRAF^wt^ (including NRAS^mut^) melanoma, no efficient TT is currently available and 40–60% of patients do not respond to immune checkpoint inhibition due to intrinsic resistance [32]. MEKi monotherapy has shown some clinical activity with a very short duration of response before resistance emerges [33]. Therefore, attempts were undertaken with combinational strategies supplementing MEKi, which have not improved the outcome so far. For example, the MEK inhibitor cobimetinib combined with the PD-L1 antibody atezolizumab did not improve PFS compared to monotherapy with the PD-1 antibody pembrolizumab [34]. However, combinations of MEKi with BRAFi produce anti-tumor effects in NRAS^mut^ melanoma cells by inducing MAPK pathway-independent endoplasmic reticulum (ER) stress [35], indicating that BRAFi/MEKi combinations may be an effective and well-tolerated treatment option for patients with NRAS^mut^ melanoma.

Until now, no direct side-by-side comparison was conducted for BRAF^mut^ melanoma for the three approved BRAFi/MEKi combinations in vitro, in vivo, or in any clinical trials. Existing data of indirect comparisons suggested similar efficacies with marginal differences in the median overall survival (33.6 months: encorafenib/binimetinib, 25.3 months: dabrafenib/trametinib, and 22.3 months: vemurafenib/cobimetinib) [3,36,37,38]. Therefore, our study aimed to close this gap by investigating the anti-proliferative, pro-apoptotic and resistance-inducing effects of approved and not-approved BRAFi/MEKi combinations in both BRAF^mut^ and NRAS^mut^ melanoma cells.

## 2. Materials and Methods

### 2.1. Cell Culture and Reagents

Melanoma cell lines Malme3M (lung metastasis, BRAF^mut^), WM3734 (brain metastasis, BRAF^mut^), and WM1366 (primary melanoma, vertical growth phase, NRAS^mut^) were purchased from ATCC (Manassas, VI, USA) and Rockland Immunochemicals, Inc. (Pottstown, PA, USA) and were subject to regular tests for excluding Mycoplasma contamination. Malme3M was cultured in RPMI + 20% fetal bovine serum (FBS), WM3734 and WM1366 were cultured in RPMI + 10% FBS. Encorafenib, binimetinib, vemurafenib, cobimetinib, dabrafenib, and trametinib were purchased from Selleckchem and dissolved in dimethylsulfoxide (DMSO).

### 2.2. Analysis of Cell Cycle and Apoptosis (SubG1-Fraction) by Propidium-Iodide Staining and Flow Cytometry

Cells were seeded at 2 × 10^5^ cells (Malme3M) or 1.75 × 10^5^ cells (WM3734 and WM1366) per well in 6-well plates and incubated for 24 h at 37 °C. Following this incubation, cells were treated with serial dilutions of 9 different BRAFi and MEKi combinations (BRAF^mut^ cells: ratio 10:1 as previously described [39,40,41,42,43,44], range 0.003/0.0003–10/1 µM, 1:8 dilutions, NRAS^mut^ cells: ratio 10:1, range 0.3125/0.03125–10/1 µM, 1:2 dilutions) or a DMSO control, corresponding to the highest volume of inhibitor added in new growth medium. After incubation for 72 h, floating and adherent cells were collected and centrifuged at 300× *g* for 5 min at 4 °C. Cell pellets were washed twice with PBS and centrifuged again at 300× *g* for 5 min at 4 °C. Cells were then fixed by adding 75% EtOH dropwise while gently vortexed to avoid clumping of the cells. Following fixation overnight at 4 °C, cells were washed twice with PBS and centrifuged at 500× *g* for 5 min at 4 °C. Cells were then incubated with propidium-iodide solution (25 µg/mL of propidium-iodide and 100 µg/mL of RNAse A in PBS) at room temperature for 20 min. Finally, an analysis of the cell cycle for the percentage of subG1 fraction (apoptotic fraction) by flow cytometry (Fortessa, BD) was conducted with subsequent software analysis using Diva and FlowJo. Data were plotted using GraphPad Prism 7 software. Significance was defined as *p* < 0.05 based on a two-tailed non-paired *t*-test.

### 2.3. Proliferation Assay with MUH-Reagent

Cells were seeded at 7000 cells (Malme3M), 5500 cells (WM3734), or 4500 cells (WM1366) per well in 96-well plates and incubated for 24 h. Following this incubation, cells were treated with BRAFi and/or MEKi or DMSO control for 72 h, as indicated above. Following two PBS washes, cells were incubated with 100 µL of a solution containing 100 µg/mL 4-methylumbelliferyl-heptanoate (MUH) in PBS for 1 h at 37 °C. The absolute fluorescence intensity at λ_ex_ of 355 nm and λ_em_ of 460 nm was measured using a fluorescent plate reader (TECAN). The intensity of fluorescence corresponds to the number of viable cells. The percentage of proliferating cells in each sample was calculated by normalization to the DMSO-treated control. Data were plotted using Microsoft Excel.

### 2.4. Calculation of IC_50_

The relative half-maximal inhibitory concentrations (IC_50_) for single and combined BRAFi/MEKi treatment regimens were calculated from the normalized MUH proliferation data using GraphPad Prism 7. For this, the curve-fitting function “non-linear regression” and the option “log(inhibitor) vs. response—Variable slope (four parameters)” were selected, constraining the bottom plateau to be a constant equal to the maximal inhibition value achieved by one of the inhibitor combinations. The highest concentration from all single and combined dabrafenib treatment regimens was omitted, to exclude the “off-target” effects of dabrafenib from the IC_50_ calculations. Resulting IC_50_ curves were plotted onto mean values ± SD of normalized MUH proliferation data using GraphPad Prism 7 software. Significance was defined as *p* < 0.05 based on a two-tailed non-paired *t*-test.

### 2.5. Calculation of Synergy Scores

For determining the synergistic potential of all inhibitor combinations, normalized MUH proliferation data were transformed to represent the percentage of inhibition instead of percentage proliferation by subtracting the normalized values from 100 percent of DMSO control. Next, data were uploaded to SynergyFinder 3.0 [45] and option “LL4” was chosen for curve fitting, and synergy scores were calculated by using the “Bliss” method. Synergy scores below −10 are indicative of antagonistic effects, values between −10 and 10 suggest additive effects, and values of more than 10 represent synergistic effects.

### 2.6. Creating Resistant Cell Lines

Resistant cells were generated by continuously treating cell lines with doubling concentrations of BRAFi/MEKi combinations. The medium was renewed twice a week including the addition of fresh inhibitors. Inhibitor concentrations were raised when cell lines were able to tolerate the previous dose by resuming proliferation. The selection criteria for choosing the maximum concentration of inhibitors to create fully resistant cell lines were the following: (i) the concentration causes >60% inhibition of proliferation in all three tested cell lines, (ii) the concentration must be similar or preferably below the maximum concentration of inhibitors reported in the plasma of treated patients.

## 3. Results

### 3.1. Approved BRAFi/MEKi Combinations Vary in Their Efficacy to Induce Apoptosis

In order to assess the efficacy of all possible BRAFi/MEKi combinations, three melanoma cell lines were treated using nine combinations (encorafenib/binimetinib; encorafenib/cobimetinib; encorafenib/trametinib; vemurafenib/binimetinib; vemurafenib/cobimetinib; vemurafenib/trametinib; dabrafenib/binimetinib; dabrafenib/cobimetinib; dabrafenib/trametinib) with increasing concentrations for 72 h. The apoptotic (subG1) fraction was subsequently analyzed by the flow cytometry of permeabilized, propidium-iodide-stained cells. In all three melanoma cell lines, the percentage of apoptotic cells was lowest when treated with the BRAFi vemurafenib (blue) compared to encorafenib (green) or dabrafenib (magenta), especially at lower concentrations (Figure 1, Appendix A). Although both encorafenib and dabrafenib displayed similar rates of apoptosis, a minimal advantage of encorafenib over dabrafenib was observed in both BRAF^mut^ cell lines (left and middle panel, serial dilutions 1:8). Notably, the NRAS^mut^ cell line WM1366 showed similar rates of apoptosis as the BRAF^mut^ cell lines, with the exception of vemurafenib/binimetinib treatment which remained ineffective up to the maximum concentration of 10 µM of vemurafenib and 1 µM of binimetinib (right panel, serial dilutions 1:2).

Regarding the efficacy of MEKi in the combined BRAFi/MEKi setting, apoptosis rates were highest for trametinib (magenta), followed by cobimetinib (blue) and binimetinib (green) (Figure 2, Appendix A). This effect was more pronounced at lower concentrations of inhibitors, especially in the NRAS^mut^ cell line WM1366. In all cell lines, trametinib induced the highest rates of apoptosis irrespective of the BRAFi used. Conversely, binimetinib treatment combined with any of the BRAFi showed apoptotic effects only at higher concentrations. The curves comparing the efficacy of MEKi for all cell lines were clearly separated when the BRAFi vemurafenib was applied (middle row). Of note, apoptotic rates induced by BRAFi/MEKi in normal human fibroblasts, melanocytes, and keratinocytes were below 10% [35].

In summary, when comparing only the three approved inhibitor combinations, dabrafenib/trametinib emerged as the best combination (Appendix A, Appendix A). However, among all nine possible combinations, encorafenib/trametinib exhibited the marginally best pro-apoptotic activity, especially at lower concentrations (Appendix A).

### 3.2. Combined BRAFi/MEKi Vary in Their Efficacy to Inhibit Proliferation and Act Synergistically Only in the NRAS^mut^ Cell Line

In addition to investigating the pro-apoptotic efficacy of all BRAFi/MEKi combinations, the anti-proliferative potential in suppressing melanoma growth in vitro was also determined. Of note, only minimal anti-proliferative effects of BRAFi and MEKi were observed in normal human cells such as fibroblasts, melanocytes, and keratinocytes [35]. In contrast, in the BRAF^mut^ cell line Malme3M, proliferation was inhibited by both monotherapies as well as combined treatment, with the “stronger” inhibitor of the combination driving the anti-proliferative activity (Figure 3). For instance, when employing encorafenib as selective BRAFi, additional anti-proliferative effects by the MEKi were marginal. On the other hand, in vemurafenib-treated cells, the addition of any MEKi caused increased inhibition of proliferation over vemurafenib alone. Of note, for higher concentrations of the BRAFi dabrafenib, growth inhibition was not as potent, indicating off-target effects of this BRAFi [46] that may occur at concentrations far beyond the serum level. Very similar data were obtained for the BRAF^mut^ WM3734 cells, although this cell line generally demonstrated reduced sensitivity compared to Malme3M with all inhibitors (Figure 4).

As expected, the NRAS^mut^ cell line WM1366 barely reacted to any of the three single BRAFi (Figure 5). Additionally, while single MEKi caused only moderate anti-proliferative activity, the combination with BRAFi greatly augmented these effects, acting in a synergistic manner (see discussion for further detail). The extent of inhibition depended on the BRAFi tested, with vemurafenib generally showing the lowest activity. The “off-target” effects of dabrafenib at higher concentrations were also visible in this NRAS^mut^ cell line.

Taken together, proliferation data suggested that the combination of vemurafenib with binimetinib was least effective to inhibit melanoma growth in vitro compared with all other combinations. On the other hand, inhibitor combinations such as encorafenib/trametinib and encorafenib/cobimetinib showed very promising inhibitory actions. Importantly, both of these inhibitor combinations are currently not used in melanoma treatment or investigated in any clinical trials. For the approved inhibitor combinations, dabrafenib/trametinib provided the best anti-proliferative activity.

### 3.3. Resistance to BRAFi/MEKi Depended on the Used Combination and Mutation Status

Irrespective of the clinically approved BRAFi/MEKi combination applied (for BRAF^mut^ melanoma: encorafenib/binimetinib; vemurafenib/cobimetinib; dabrafenib/trametinib), melanoma cells will eventually develop resistance to this type of treatment. In order to investigate the time to develop resistance, two BRAF^mut^ and one NRAS^mut^ cell line(s) were subjected to continuous treatment with doubling concentrations of clinically approved BRAFi/MEKi combinations. Dabrafenib/trametinib treatment resulted in the development of complete resistance after a period of 15.1 months for Malme3M and 16.1 months for WM3734 (both BRAF^mut^, Table 1). Vemurafenib/cobimetinib caused complete resistance after 14.6 months for Malme3M and 19.3 months for WM3734 cells. Time to complete resistance to the maximum concentration was the longest for encorafenib/binimetinib, emerging after 31.5 months for Malme3M and 21.0 months for WM3734. Notably, WM3734 cells resistant to encorafineb/binimetinib exhibited a quiescent and slow cycling phenotype that was insensitive to further increases in the concentration of BRAFi/MEKi. In contrast to BRAF^mut^ cell lines, the NRAS^mut^ cell line WM1366 developed resistance after exposing cells to continuous BRAFi/MEKi treatment in a much shorter time frame, in only 2.4 months (encorafenib/binimetinib), 3.2 months (vemurafenib/cobimetinib), and 5.1 months (dabrafenib/trametinib).

## 4. Discussion

The currently approved BRAFi/MEKi combinations have demonstrated significant initial response rates in patients suffering from BRAF^mut^ metastatic melanoma. However, the majority of patients develop drug resistance, and a direct side-by-side comparison of these three inhibitor combinations is still lacking. In this study, we tested the approved BRAFi/MEKi combinations encorafenib/binimetinib, vemurafenib/cobimetinib, and dabrafenib/trametinib, as well as not-approved BRAFi/MEKi in a direct side-by-side comparison in vitro. This approach aimed to identify the most efficient BRAFi/MEKi combination (i) to inhibit tumor progression and (ii) to delay the onset of resistance.

The data of this experimental study indicate that the effects of BRAFi/MEKi on BRAF^mut^ cell lines were non-synergistical, with the stronger inhibitory agent being the dominant driver for anti-tumor activity (Table 2). This observation may be explained by the fact that both inhibitors exert their effects on the same pathway. In contrast to treatment with the single inhibitors, combination treatment is thereby able to delay resistance by preventing or delaying the reactivation of the MAPK pathway in patients with BRAF^mut^ melanoma. In combination with MEKi, the least effective BRAFi was vemurafenib, and the highest anti-tumor efficacy was achieved by encorafenib. The most efficient MEKi in combination with BRAFi was trametinib, closely followed by cobimetinib, while binimetinib exhibited the lowest anti-tumor efficacy. Consequently, the most effective BRAFi/MEKi combination was encorafenib plus trametinib, the least effective combination was vemurafenib/binimetinib. These observations are supported by half-maximal inhibitory concentration (IC_50_) values calculated for all tested combinations (Appendix A). Intriguingly, the combinations with lowest IC_50_ values for combined treatment were encorafenib/trametinib, dabrafenib/trametinib, encorafenib/cobimetinib, and dabrafenib/cobimetinib in all three cell lines (Table 2, see Appendix A for single and combined inhibition), with dabrafenib/trametinib being the only approved inhibitor combination within these top four. Our results correlate well with the already published pharmacological characteristics for the individual inhibitors. As such, Delord et al. showed that encorafenib exhibits the longest dissociation half-life (>30 h vs. 2 h dabrafenib vs. 0.5 h vemurafenib) and the lowest IC_50_ values for the inhibition of proliferation in a wide variety of melanoma cell lines (<0.04 µM encorafenib vs. <0.1 µM dabrafenib vs. <1 µM vemurafenib) [49]. Among MEKi, trametinib (0.7 nM MEK1, 0.9 nM MEK2) showed the lowest half-maximal inhibitory concentration (IC_50_) for MEK1 and MEK2, respectively, compared with cobimetinib (0.9 nM MEK1, 199 nM MEK2) and binimetinib (12 nM MEK1/MEK2) [50,51].

In contrast to BRAF^mut^ cells, BRAFi/MEKi appear to act synergistically in the NRAS^mut^/BRAF^wt^ cell line WM1366. This is perhaps not surprising, as in the absence of the BRAF mutation in these cells, BRAFi act on pathways different from MAPK, thereby exerting additional cellular toxicity, for example, through ER stress [35,52]. Thus, in the combination treatment, the ER stress-inducing effects of BRAFi enhance the effects of MEKi in a synergistic manner. The highest synergy scores were calculated for encorafenib/cobimetinib, encorafenib/trametinib, and dabrafenib/cobimetinib. The fourth best synergy score was calculated for the currently approved dabrafenib/trametinib combination (Figure 6, Table 2).

BRAF^mut^ cell lines exhibited complete resistance to final concentrations of BRAFi/MEKi after 14.6 to 30.0 months, correlating well with clinical data from patients treated with approved BRAFi/MEKi combinations (Table 1). As such, encorafenib plus binimetinib was the most effective to delay the onset of complete resistance in BRAF^mut^ cell lines, most likely caused by the highly effective BRAFi. On the other hand, the same combination in NRAS^mut^ WM1366 cells led to the most rapid development of resistance, most likely because of the weak efficacy of binimetinib. The time required for NRAS^mut^ cells to develop resistance was longest when using dabrafenib/trametinib, possibly due to the strong MEK inhibition. Given that all BRAFi were designed to target mutated codon 600 of BRAF, it was not surprising that the time to the evolution of resistance to BRAFi/MEKi was generally shorter in NRAS^mut^/BRAF^wt^ cells. However, in light of limited treatment options for this subgroup of patients with a poor prognosis, treatment with BRAFi/MEKi may be beneficial and superior to MEKi monotherapy.

Even though melanoma cells initially respond to all inhibitor combinations, the emergence of resistance to any of the BRAFi/MEKi combinations will occur in the majority of patients. However, complementing the current set of tools with alternative powerful combinations could sustain the rationale for re-challenging initially responsive tumors after a progressive disease.

## 5. Conclusions

In summary, for exploiting the full potential of combined BRAF/MEK inhibition, all possible BRAFi/MEKi combinations were evaluated in three melanoma cell lines. For the approved BRAFi/MEKi combinations, the head-to-head comparison exposed vemurafenib/cobimetinib to be clearly less efficient than the other two combinations, while dabrafenib/trametinib was initially superior (Malme3M, WM1366) or comparable to encorafenib/binimetinib (WM3734). However, the encorafenib/binimetinib combination was distinctly most successful in delaying resistance in BRAF^mut^ melanoma. Furthermore, the data revealed that unconventional combinations, such as encorafenib/trametinib or encorafenib/cobimetinib, exhibited the highest efficacy in vitro and, therefore, deserve an opportunity to become part of clinical trials which could also enroll MEKi-naïve NRAS^mut^ melanoma patients.

## Figures and Tables

**Figure 1 cancers-14-04930-f001:**
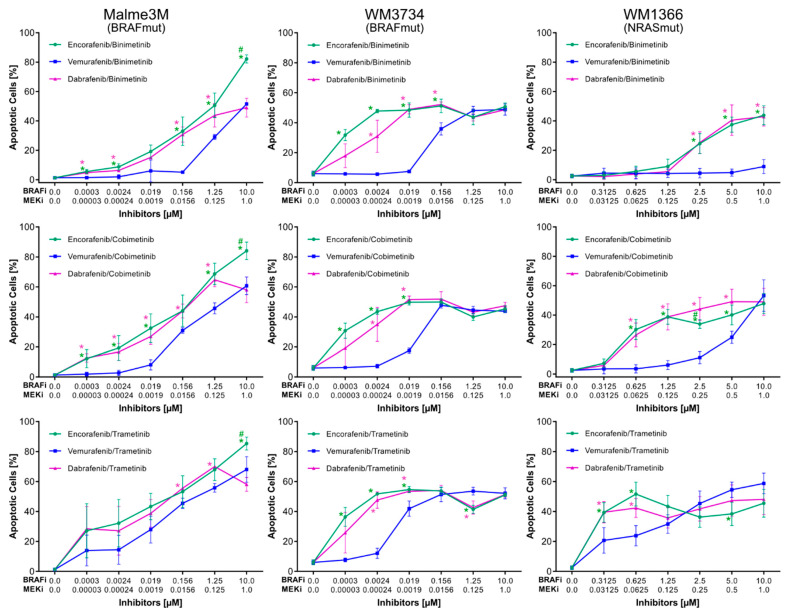
Sub-G1 analysis (apoptosis) of two BRAF^mut^ cell lines (Malme3M and WM3734) and one NRAS^mut^ cell line (WM1366) after treatment for 72 h using different BRAFi/MEKi combinations with increasing concentrations of inhibitors (BRAF^mut^ 1:8 serial dilutions, NRAS^mut^ 1:2 serial dilutions). *n* ≥ 3, error bars indicate standard deviation, * indicates significantly different values relative to vermurafenib, # labels data points significantly different relative to dabrafenib. Significance defined as *p* < 0.05 based on a two-tailed non-paired *t*-test. Appendix A shows a complete list of *p*-values for all data points.

**Figure 2 cancers-14-04930-f002:**
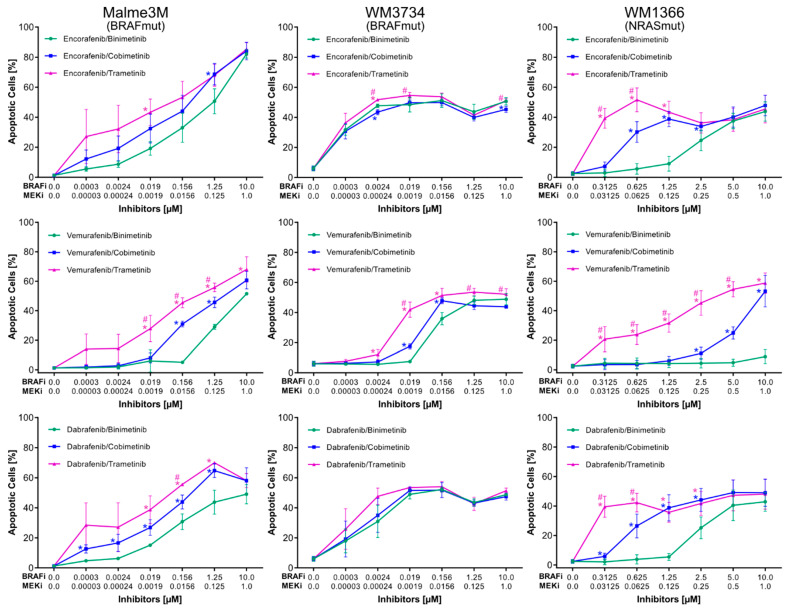
Sub-G1 (apoptosis) of two BRAF^mut^ cell lines (Malme3M and WM3734) and one NRAS^mut^ cell line (WM1366) after treatment for 72 h using different BRAFi/MEKi combinations with increasing concentrations of inhibitors (BRAF^mut^ 1:8 serial dilutions, NRAS^mut^ 1:2 serial dilutions). Identical data, as in Figure 1, are arranged differently for improved comparability; *n* ≥ 3, error bars indicate standard deviation, * indicates significantly different values relative to binimetinib, # labels data points that are significantly different relative to cobimetinib. Significance defined as *p* < 0.05 based on two-tailed non-paired *t*-test. Appendix A shows a complete list of *p*-values for all data points.

**Figure 3 cancers-14-04930-f003:**
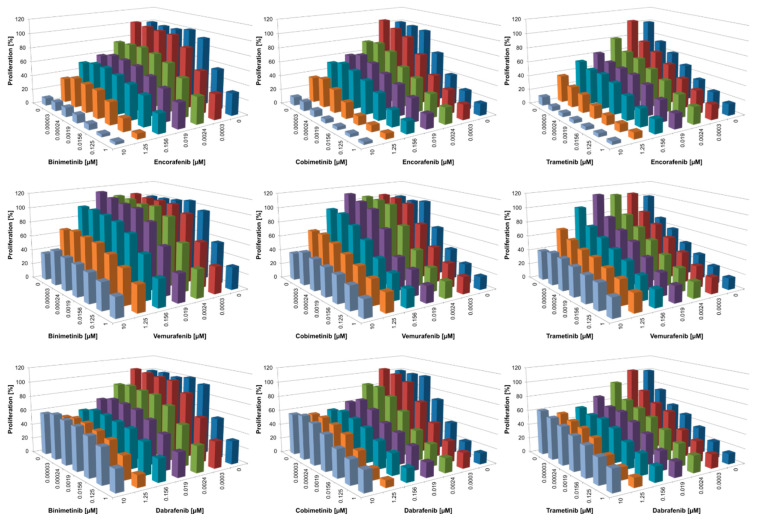
Inhibition of proliferation of Malme3M (BRAF^mut^) after treatment for 72 h with single or combined BRAFi/MEKi (serial dilutions 1:8); *n* ≥ 3; for simplicity, standard deviation and significance are not shown. Significance from these data is displayed in Appendix A. List of *p*-values can be found in Appendix A.

**Figure 4 cancers-14-04930-f004:**
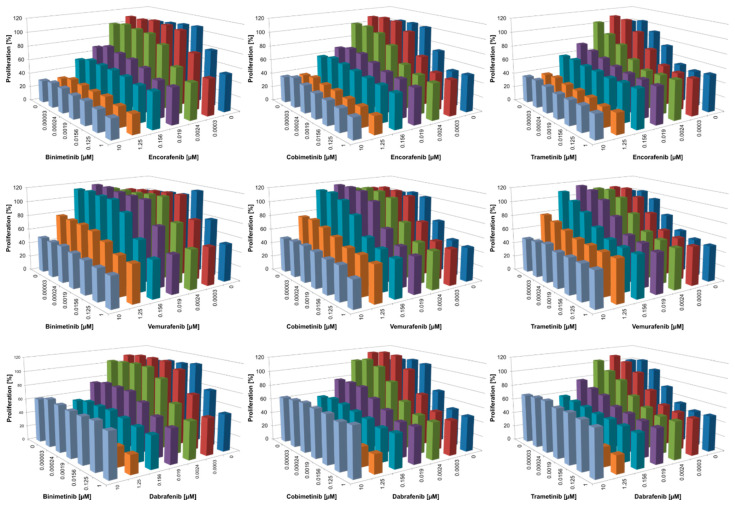
Inhibition of proliferation of WM3734 (BRAF^mut^) after treatment for 72 h with single or combined BRAFi/MEKi (serial dilutions 1:8); *n* ≥ 3; for simplicity, the standard deviation and significance are not shown. Significance from these data is displayed in Appendix A. List of *p*-values can be found in Appendix A.

**Figure 5 cancers-14-04930-f005:**
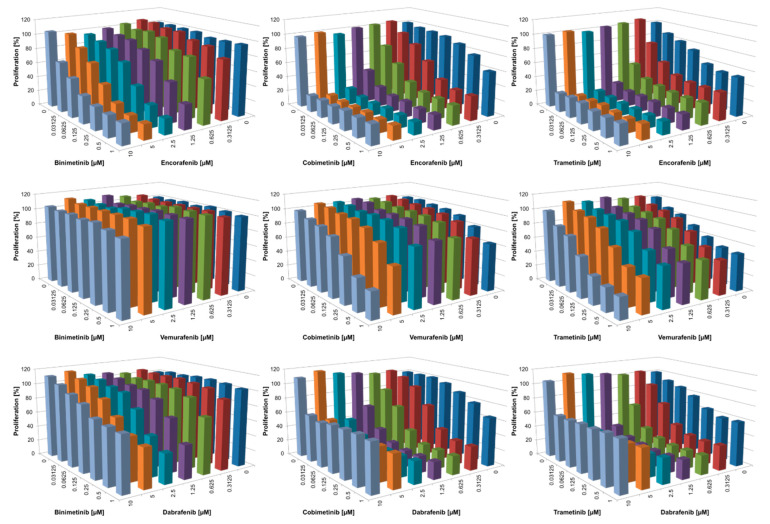
Inhibition of proliferation of WM1366 (NRAS^mut^) after treatment for 72 h with single or combined BRAFi/MEKi (serial dilutions 1:2); *n* ≥ 3; for simplicity, standard deviation and significance are not shown. Significance from these data is displayed in Appendix A. List of *p*-values can be found in Appendix A.

**Figure 6 cancers-14-04930-f006:**
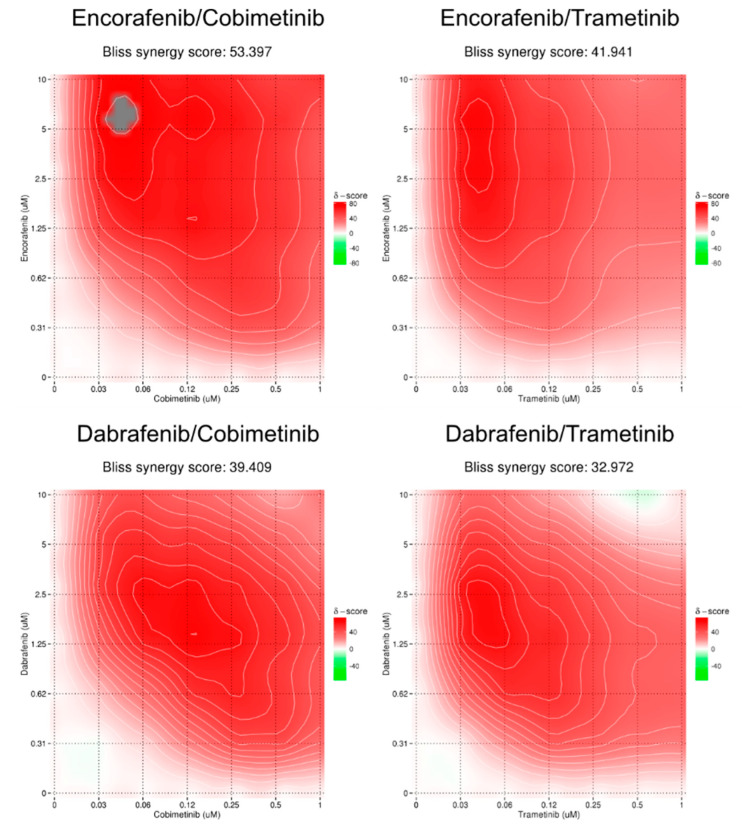
Synergy maps of the top four synergistic combinations in WM1366 (NRAS^mut^) cell line after treatment for 72 h with single or combined BRAFi/MEKi (serial dilutions 1:2).

**Table 1 cancers-14-04930-t001:** Comparison of inhibitor characteristics and efficacies in the resistant cell lines and clinical studies.

Inhibitors	Max. Plasma Concentration * In Vivo [µM]	Max. Concentration In Vitro [µM]	Time to Resistance In Vitro [months]	Median PFS In Vivo [months]	DOR In Vivo [months]	Study
Encorafenib	7.04	2.5	**31.5** (Malme3M) **21.0** (WM3734) 2.4 (WM1366)	14.9	18.6	[36,38,47]
Binimetinib	1.48	0.25				
Vemurafenib	116	10	14.6 (Malme3M) 19.3 (WM3734) 3.2 (WM1366)	12.3	13.0	[36,38,48]
Cobimetinib	0.51	0.5				
Dabrafenib	2.84	0.625	15.1 (Malme3M) 16.1 (WM3734) **5.1** (WM1366)	11.4	13.8	[30,36,38]
Trametinib	0.036	0.0625				

PFS—progression-free survival; DOR—duration of response; bold numbers represent the best combination for each cell line. ***** calculated from the molecular weight and maximum plasma concentration (C_max_ in ng/mL), which was obtained from the product information sheet at the European Medicines Agency (https://www.ema.europa.eu/en, accessed on 4 August 2022) or from Delord et al., 2017 [49]. Note that the volume of distribution for trametinib is 1200 L, indicating a much higher concentration of trametinib in the tissue compared to the serum.

**Table 2 cancers-14-04930-t002:** Comparison of synergy score and IC_50_ data of the different BRAFi and MEKi combinations.

**Synergy Score**									
	**E/B**	**E/C**	**E/T**	**V/B**	**V/C**	**V/T**	**D/B**	**D/C**	**D/T**
**Malme3M**	−2.493	−3.802	−3.416	−4.191	−7.428	−12.375	−3.824	−5.877	−8.547
**WM3734**	−3.251	−4.270	−5.341	−2.740	−4.603	−7.209	−3.383	−5.252	−6.064
**WM1366**	28.262	53.397	41.941	−3.484	−2.710	−7.199	12.137	39.409	32.972
**IC_50_ (µM)**									
	**E/B**	**E/C**	**E/T**	**V/B**	**V/C**	**V/T**	**D/B**	**D/C**	**D/T**
**Malme3M**	0.0537	0.0194	0.0107	1.0780	0.1684	0.0620	0.1294	0.0227	0.0119
	0.00537	0.00194	0.00107	0.10780	0.01684	0.00620	0.01294	0.00227	0.00119
**Ranking**	5	3	1	9	8	6	7	4	2
**WM3734**	0.0356	0.0247	0.0143	0.8345	0.2885	0.3022	0.0458	0.0271	0.0145
	0.00356	0.00247	0.00143	0.08345	0.02885	0.03022	0.00458	0.00271	0.00145
**Ranking**	5	3	1	9	7	8	6	4	2
**WM1366**	1.9830	0.5985	0.4139	NC	6.9750	4.6110	4.6610	0.6812	0.4631
	0.19830	0.05985	0.04139	NC	0.69750	0.46110	0.46610	0.06812	0.04631
**Ranking**	5	3	1	9	8	6	7	4	2

IC_50_—half-maximal inhibitory concentration; E—encorafenib; V—vemurafenib; D—dabrafenib; B—binimetinib; C—cobimetinib; T—trametinib; NC—not converged; red numbers indicate the four best-performing combinations.

## Data Availability

Data is contained within the article and Appendix A.

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
