# Peer review of "Head-to-Head Comparison of BRAF/MEK Inhibitor Combinations Proposes Superiority of Encorafenib Plus Trametinib in Melanoma"

_cancers, 2022, doi:10.3390/cancers14194930_

Round 1

Reviewer 1 Report

In this submission, Schulz and colleagues examined several combination therapies of BRAF/MEK inhibitors to help find therapeutic solutions for the hard-to-treat NRAS mutations in melanoma. It has been shown that there are several mechanisms of acquired resistance have emerged which require the development of new models of combination targeted therapies.

I have some comments to be addressed: 

Q1: why the authors decided to perform a cell cytotoxicity assay using methylumbelliferyl-heptanoate (MUH) where the MTT and Alamar blue are highly used in such applications?

Q2: I assume that the authors believe that these compounds are not toxic to regular cell lines. I think a control cell line should be included in this study, such as normal lymphocytes isolated from healthy donors or hTERT cells. The goal is to identify the combinations that target cancer cells, not normal healthy cells.

 Q3: Why is the ratio of BRAFi to MEKi 10:1? What is the reason for selecting this ratio?

Reviewer 2 Report

The manuscript was well organized  and written. The experiments were well done and presented. What make me very confused is "the effects of BRAFi/MEKi on BRAFmut cell lines were non-synergistically with the stronger inhibitory agent being the dominant driver for anti-tumor activity". How? Does it meant that higher dose of BRAFi or MEKi will achieve the same effect of BRAFi/MEKi combination?  "In contrast to BRAFmut cells, BRAFi/MEKi appear to act synergistically on NRASmut/BRAFwt cell line WM1366, indicating that BRAFi in these cells may act on pathways different from MAPK, thereby exerting additional cellular toxicity" How? Why does the same recombination function differently in different cancer cells? It will be of great help to do some WB  to elucidate the underlying altered molecular pathways after the use of inhibitors. 
